# RECURRENT KALMAN NETWORKS:
# FACTORIZED INFERENCE IN HIGH-DIMENSIONAL DEEP FEATURE SPACES

## ABSTRACT

In order to integrate uncertainty estimates into deep time-series modelling, Kalman Filters (KFs) (Kalman et al., 1960) have been integrated with deep learning models. Yet, such approaches typically rely on approximate inference techniques such as variational inference which makes learning more complex and often less scalable due to approximation errors. We propose a new deep approach to Kalman filtering which can be learned directly in an end-to-end manner using backpropagation without additional approximations. Our approach uses a high-dimensional factorized latent state representation for which the Kalman updates simplify to scalar operations and thus avoids hard to backpropagate, computationally heavy and potentially unstable matrix inversions. Moreover, we use locally linear dynamic models to efficiently propagate the latent state to the next time step. While our locally linear modelling and factorization assumptions are in general not true for the original low-dimensional state space of the system, the network finds a high-dimensional latent space where these assumptions hold to perform efficient inference. This state representation is learned jointly with the transition and noise models. The resulting network architecture, which we call *Recurrent Kalman Network* (RKN), can be used for any time-series data, similar to a LSTM (Hochreiter and Schmidhuber, 1997) but uses an explicit representation of uncertainty. As shown by our experiments, the RKN obtains much more accurate uncertainty estimates than an LSTM or Gated Recurrent Units (GRUs) (Cho et al., 2014) while also showing a slightly improved prediction performance and outperforms various recent generative models on an image imputation task.

## 1    INTRODUCTION

State-estimation in unstructured environments is a very challenging task as observations or measurements of the environment are often high-dimensional and only provide partial information about the state. Images are a good example: Even for low resolution, the number of pixels can quickly exceed tens or hundreds of thousands and it is impossible to obtain any information about the dynamics, such as velocities, from a single image. Additionally, the observations may be noisy or may not contain useful information for the task at hand. Such noise can for example be introduced by poor illumination or motion blur and occlusions can prevent us from observing some or all relevant aspects of the scene. In addition to state estimation, it is also often desirable to predict future states or observations, for example, in order to assess the consequences of future actions. To this end, an initial estimate of the current state is necessary which again has to be inferred from observations. In such environments we typically also have to deal with a high uncertainties in the state estimates. Being able to model this uncertainty is crucial in many decision making scenarios, e.g., if we need to decide to perform an action now or wait until more information about the scene is available.

Deep learning models have been very successful for time-series modelling in unstructured environments. Classical models such as LSTMs (Hochreiter and Schmidhuber, 1997) or GRUs (Cho et al., 2014) perform well but fail to capture the uncertainty of the state estimate. Recent probabilistic deep learning approaches have used the Kalman filter (KF) as a tool to integrate uncertainty estimates into deep time-series modelling (Haarnoja et al., 2016; Watter et al., 2015; Archer et al., 2015; Fraccaro et al., 2017; Krishnan et al., 2017). These approaches use the KF to perform inference in

a low-dimensional (latent) state space that is typically defined by a deep encoder. However, using KF in such a state space comes with two main limitations. In order to be usable for non-linear dynamics, we have to introduce approximations such as the extended KF (Haarnoja et al., 2016) and variational inference methods (Krishnan et al., 2017; Fraccaro et al., 2017). Moreover, the KF equations require computationally expensive matrix inversions that are hard to scale to high dimensional latent spaces for more complex systems and computationally demanding to fully backpropagate in an end-to-end manner. Most of these methods are implemented as (variational) auto-encoders and are therefore also limited to predicting future observations or imputing missing observations and can not be directly be applied to state estimation.

We introduce the *Recurrent Kalman Network*, an end-to-end learning approach for Kalman filtering and prediction. While Kalman filtering in the original state space requires approximations due to the non-linear models as well as matrix inversions that are hard to back-propagate and computationally expensive, the RKN uses a learned high-dimensional latent state representation that allows for efficient inference using locally linear transition models and a factorized belief state representation which avoids expensive and numerically problematic matrix inversions. Conceptually, this idea is related to kernel methods which use high-dimensional feature spaces to approximate nonlinear functions with linear models (Gebhardt et al., 2017). However, in difference to kernel feature spaces, our feature space is given by a deep encoder and learned in an end-to-end manner.

The RKN can be used for any time-series data set for which LSTMs and GRUs are currently the state of the art. In contrast to those, the RKN uses an explicit representation of uncertainty which governs the gating between keeping the current information in memory or updating it with the current observation. While the RKN shows a slightly improved performance in terms of state estimation errors, both LSTMs and GRUs struggle with estimating the uncertainty of the prediction while the RKN can provide accurate uncertainty estimates. In relation to existing KF-based approaches, our approach can be used for state estimation as well as for generative tasks such as image imputation. We also show that we outperform state of the art methods on a complex image imputation task.

## 1.1 RELATED WORK

Using encoders for time-series modelling of high-dimensional data such as images is a common approach. Such encoders can also be easily integrated in well known deep time-series models such as LSTMs (Hochreiter and Schmidhuber, 1997) or GRUs (Cho et al., 2014). These models are very effective but do not provide good uncertainty estimates as shown in our experiments. Pixel to Torques (P2T) (Wahlström et al., 2015) employs an autoencoder to obtain low dimensional latent representations from images together with a transition model. They subsequently use the models to perform control in the latent space. Embed to Control (E2C) (Watter et al., 2015) can be seen as an extension of the previous approach with the difference that a variational autoencoder (Kingma and Welling, 2013) is used. However, both of these approaches are not recurrent and rely on observations which allow inferring the whole state from a single observation. They can therefore not deal with noisy or missing data.

Another family of approaches interprets encoder-decoder models as latent variable models that can be optimized efficiently by variational inference. They derive a corresponding lower bound and optimize it using the stochastic gradient variational Bayes approach (Kingma and Welling, 2013). Black Box Variational Inference (BB-VI) (Archer et al., 2015) proposes a structured Gaussian variational approximation of the posterior, which simplifies the inference step at the cost of maintaining a tri-diagonal covariance matrix of the full state. To circumvent this issue, Structured Inference Networks (SIN) (Krishnan et al., 2017) employ a flexible recurrent neural network to approximate the dynamic state update. Deep Variational Bayes Filters (DVBF) (Karl et al., 2016) integrate general Bayes filters into deep feature spaces while the Kalman Variational Autoencoder (KVAE) (Fraccaro et al., 2017) employs the classical Kalman Filter and allows not only filtering but also smoothing.

Variational Sequential Monte Carlo (VSMC) (Naesseth et al., 2017) uses particle filters instead, however they are only learning the proposal function and are not working in learned latent spaces. Yet, these models can not be directly used for state estimation as they are formulated as generative models of the observations without the notion of the real state of the system. Moreover, the use of variational inference introduces an additional approximation that might affect the performance of the algorithms. The approaches given in (Archer et al., 2015; Fraccaro et al., 2017; Haarnoja et al.,

2016) directly use the Kalman update equations in the latent state, which limits these approaches to rather low dimensional latent states due to the expensive matrix inversions.

The BackpropKF (Haarnoja et al., 2016) applies a CNN to estimate the observable parts of the true state given the observation. Similar to our approach, this CNN additionally outputs a covariance matrix indicating the models certainty about the estimate and allows the subsequent use of an (extended) Kalman filter with known transition model. In contrast, we let our model chose the feature space that is used for the inference such that locally linear models can be learned and the KF computations can be simplified due to our factorization assumptions.

A summary of all the approaches, listing also their basic properties, can be seen in Table 1. We compare the approaches whether they are scaleable to high dimensional latent states, whether they can be

|       | scale-able | state est. | uncer-tainty | noise | direct opt. |
|-------|:---:|:---:|:---:|:---:|:---:|
| LSTM  | ✓ | ✓ | ×/✓ | ✓ | ✓ |
| GRU   | ✓ | ✓ | ×/✓ | ✓ | ✓ |
| P2T   | ✓ | ✓ | ×/✓ | × | ✓ |
| E2C   | ✓ | × | ✓ | × | × |
| BB-VI | × | × | ✓ | ✓ | × |
| SIN   | ✓ | × | ✓ | ✓ | × |
| KVAE  | × | × | ✓ | ✓ | × |
| DVBF  | ✓ | × | ✓ | ✓ | × |
| VSMC  | ✓ | × | ✓ | ✓ | × |
| **RKN** | ✓ | ✓ | ✓ | ✓ | ✓ |

Table 1: Qualitative comparison of our approach to recent related work.

used for state estimation, whether they can provide uncertainty estimates, whether they can handle noise or missing data and whether the objective can be optimized directly or via a lower bound. All probabilistic generative models rely on variational inference which optimizes a lower bound, which potentially affects the performance of the algorithms. The P2T and E2C approaches rely on the Markov assumption and therefore can not deal with noise (or need very large window sizes). Traditional recurrent models (LSTMs, GRUs) can be trained directly by backpropagation through time and therefore typically yield very good performance but are lacking uncertainty estimates (which, however, can be artificially added as in our experiments). Our RKN approach combines the advantages of all methods above as it can be learned by direct optimization without the use of a lower bound and it provides a principled way of representing uncertainty inside the neural network.

## 2 FACTORIZED INFERENCE IN DEEP FEATURE SPACES

Lifting the original input space to a high-dimensional feature space where linear operations are feasible is a common approach in machine learning, e.g., in kernel regression and SVMs. The *Recurrent Kalman Network* (RKN) transfers this idea to state estimation and filtering, i.e., we learn a high dimensional deep feature space that allows for efficient inference using the Kalman update equations even for complex systems with high dimensional observations. To achieve this, we assume that the belief state representation can be factorized into independent Gaussian distributions as described in the following sections.

### 2.1 LATENT OBSERVATION AND STATE SPACES

The RKN encoder learns a mapping to a high-dimensional latent observation space $\mathcal{W} = \mathbb{R}^m$. The encoder also outputs a vector of uncertainty estimates $\boldsymbol{\sigma}_t^{\text{obs}}$, one for each entry of the latent observation $\mathbf{w}_t$. Hence, the encoder can be represented by $(\mathbf{w}_t, \boldsymbol{\sigma}_t^{\text{obs}}) = \text{enc}(\mathbf{o}_t)$. The latent state space $\mathcal{Z} = \mathbb{R}^n$ of the RKN is related to the observation space $\mathcal{W}$ by the linear latent observation model $\mathbf{H} = \begin{bmatrix} \mathbf{I}_m & \mathbf{0}_{m \times (n-m)} \end{bmatrix}$, i.e., $\mathbf{w} = \mathbf{Hz}$ with $\mathbf{w} \in \mathcal{W}$ and $\mathbf{z} \in \mathcal{Z}$. $\mathbf{I}_m$ denotes the $m \times m$ identity matrix and $\mathbf{0}_{m \times (n-m)}$ denotes a $m \times (n-m)$ matrix filled with zeros.

The idea behind this choice is to split the latent state vector $\mathbf{z}_t$ into two parts, a vector $\mathbf{p}_t$ for holding information that can directly be extracted from the observations and a vector $\mathbf{m}_t$ to store information inferred over time, e.g., velocities. We refer to the former as the observation or upper part and the latter as the memory or lower part of the latent state. For an ordinary dynamical system and images as observations the former may correspond to positions while the latter corresponds to velocities.

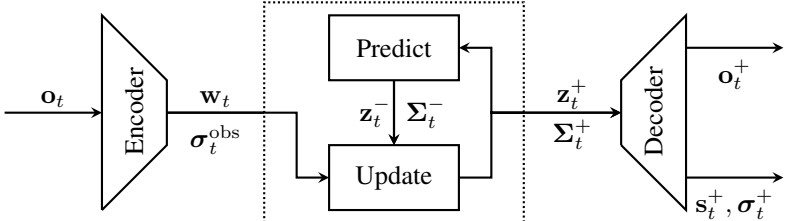

Figure 1: The *Recurrent Kalman Network*. An encoder network extracts latent features $\mathbf{w}_t$ from the current observation $\mathbf{o}_t$. Additionally, it emits an estimate of the uncertainty in the features via the variance $\boldsymbol{\sigma}_t^{\text{obs}}$. The transition model $\mathbf{A}_t$ is used to predict the current latent prior $\left(\mathbf{z}_t^-, \boldsymbol{\Sigma}_t^-\right)$ using the last posterior $\left(\mathbf{z}_{t-1}^+, \boldsymbol{\Sigma}_{t-1}^+\right)$ and subsequently update the prior using the latent observation $(\mathbf{w}_t, \boldsymbol{\sigma}_t^{\text{obs}})$. As we use a factorized representation of $\boldsymbol{\Sigma}_t$, the Kalman update simplifies to scalar operations. The current latent state $\mathbf{z}_t$ consists of the observable units $\mathbf{p}_t$ as well as the corresponding memory units $\mathbf{m}_t$. Finally, a decoder produces either $\left(\mathbf{s}_t^+, \boldsymbol{\sigma}_t^+\right)$, a low dimensional observation and an element-wise uncertainty estimate, or $\mathbf{o}_t^+$, a noise free image.

Clearly, this choice only makes sense for $m \leq n$ and in this work we assume $n = 2m$, i.e., for each observation unit $p_i$, we also represent a memory unit $m_i$ that stores its velocity information.

For each sequence, we initialize $\mathbf{z}_0^+$ with an all zeros vector and $\boldsymbol{\Sigma}_0^+$ with $10 \cdot \mathbf{I}$. In practice, it is beneficial to normalize $\mathbf{w}_t$ since the statistics of noisy and noise free images differ considerably.

## 2.2 THE TRANSITION MODEL

To obtain a locally linear transition model we learn $K$ constant transition matrices $\mathbf{A}^{(k)}$ and combine them using state dependent coefficients $\alpha^{(k)}(\mathbf{z_t})$, i.e., $\mathbf{A}_t = \sum_{k=0}^{K} \alpha^{(k)}(\mathbf{z_t})\mathbf{A}^{(k)}$. A small neural network with softmax output is used to learn $\alpha^{(k)}$. Similar approaches are used in (Fraccaro et al., 2017; Karl et al., 2016).

Using a dense transition matrix in high-dimensional latent spaces is not feasible as it contains too many parameters and causes numerical instabilities and overfitting, as preliminary experiments showed. Therefore, we design each $\mathbf{A}^{(k)}$ to consist of four band matrices $\mathbf{A}^{(k)} = [\mathbf{B}_{11}^{(k)}, \mathbf{B}_{12}^{(k)}; \mathbf{B}_{21}^{(k)}, \mathbf{B}_{22}^{(k)}]$ with bandwidth $b$. This choice reduces the number of parameters while not affecting performance since the network is free to choose the state representation.

We assume the covariance of the transition noise to be diagonal and denote the vector containing the diagonal values by $\boldsymbol{\sigma}^{\text{trans}}$. The noise is learned and independent of the state. Moreover, it is crucial to correctly initialize the transition matrix. Initially, the transition model should focus on copying the encoder output so that the encoder can learn how to extract good features if observations are available and useful. On the other hand it is crucial that $\mathbf{A}$ does not yield an instable system. We choose $\mathbf{B}_{11}^{(k)} = \mathbf{B}_{22}^{(k)} = \mathbf{I}$ and $\mathbf{B}_{12}^{(k)} = -\mathbf{B}_{21}^{(k)} = 0.2 \cdot \mathbf{I}$.

## 2.3 FACTORIZED COVARIANCE REPRESENTATION

Since the RKN learns high-dimensional representations, we can not work with the full state covariance matrices $\boldsymbol{\Sigma}_t^+$ and $\boldsymbol{\Sigma}_t^-$. We can also not fully factorize the state covariances by diagonal matrices as this neglects the correlation between the memory and the observation parts. As the memory part is excluded from the observation model $\mathbf{H}$, the Kalman update step would not change the memory nodes nor their uncertainty if we would only use a diagonal covariance matrix $\boldsymbol{\Sigma}_t^+$. Hence, for each observation node $p_i$, we compute the covariance with its corresponding memory node $m_i$. All the other covariances are neglected. This might be a crude approximation for many systems, however, as our network is free to choose its own state representation it can find a representation where such a factorization works well in practice. Thus, we use matrices of the form $\boldsymbol{\Sigma}_t = [\boldsymbol{\Sigma}_t^{\text{u}}, \boldsymbol{\Sigma}_t^{\text{s}}; \boldsymbol{\Sigma}_t^{\text{s}}, \boldsymbol{\Sigma}_t^{\text{l}}]$, where each of $\boldsymbol{\Sigma}_t^{\text{u}}, \boldsymbol{\Sigma}_t^{\text{s}}, \boldsymbol{\Sigma}_t^{\text{l}} \in \mathbb{R}^{m \times m}$ is a diagonal matrix. Again, we denote the vectors containing the diagonal values by $\boldsymbol{\sigma}_t^{\text{u}}, \boldsymbol{\sigma}_t^{\text{l}}$ and $\boldsymbol{\sigma}_t^{\text{s}}$.

### 2.4 FACTORIZED INFERENCE IN THE LATENT SPACE

Inference in the latent state space can now be implemented, similar to a standard KF, using a prediction and an observation update.

**Prediction Update.** Equivalently to the classical Kalman Filter, the current prior $\left(\mathbf{z}_t^-, \boldsymbol{\Sigma}_t^-\right)$ is obtained from the last posterior $\left(\mathbf{z}_{t-1}^+, \boldsymbol{\Sigma}_{t-1}^+\right)$ by

$$\mathbf{z}_t^- = \mathbf{A}_t \mathbf{z}_{t-1}^+ \quad \text{and} \quad \boldsymbol{\Sigma}_t^- = \mathbf{A}_t \boldsymbol{\Sigma}_{t-1}^+ \mathbf{A}_t^T + \mathbf{I} \cdot \boldsymbol{\sigma}^{\text{trans}}. \tag{1}$$

However, the special structure of the covariances enables us to significantly simplify the equation for the covariance. While being straight forward, the full derivations are rather lengthy, thus, we refer to the supplementary material where the equations are given in Eqs. 7,8 and 9.

**Observation Update.** Next, the prior is updated using the latent observation $(\mathbf{w}_t, \boldsymbol{\sigma}_t^{\text{obs}})$. Similar to the state, we split the Kalman gain matrix $\mathbf{Q}_t$ into an upper $\mathbf{Q}_t^{\text{u}}$ and a lower part $\mathbf{Q}_t^{\text{l}}$. Both $\mathbf{Q}_t^{\text{u}}$ and $\mathbf{Q}_t^{\text{l}}$ are squared matrices. Due to the simple latent observation model $\mathbf{H} = \left[ \begin{array}{cc} \mathbf{I}_m & \mathbf{0}_{m \times (n-m)} \end{array} \right]$ and the factorized covariances all off-diagonal entries of $\mathbf{Q}_t^{\text{u}}$ and $\mathbf{Q}_t^{\text{l}}$ are zero and we can work with vectors representing the diagonals, i.e., $\mathbf{q}_t^{\text{u}}$ and $\mathbf{q}_t^{\text{l}}$. Those are obtained by

$$\mathbf{q}_t^{\text{u}} = \boldsymbol{\sigma}_t^{\text{u},-} \oslash \left(\boldsymbol{\sigma}_t^{\text{u},-} + \boldsymbol{\sigma}_t^{\text{obs}}\right) \text{ and } \mathbf{q}_t^{\text{l}} = \boldsymbol{\sigma}_t^{\text{s},-} \oslash \left(\boldsymbol{\sigma}_t^{\text{u},-} + \boldsymbol{\sigma}_t^{\text{obs}}\right), \tag{2}$$

where $\oslash$ denotes an elementwise vector division. With this the update equation for the mean simplifies to

$$\mathbf{z}_t^+ = \mathbf{z}_t^- + \left[ \begin{array}{c} \mathbf{q}_t^{\text{u}} \\ \mathbf{q}_t^{\text{l}} \end{array} \right] \odot \left[ \begin{array}{c} \mathbf{w}_t - \mathbf{z}_t^{\text{u},-} \\ \mathbf{w}_t - \mathbf{z}_t^{\text{u},-} \end{array} \right] \tag{3}$$

where $\odot$ denotes the elementwise vector product. The update equations for the individual covariance parts are given by

$$\boldsymbol{\sigma}_t^{\text{u},+} = (\mathbf{1}_m - \mathbf{q}_t^{\text{u}}) \odot \boldsymbol{\sigma}_t^{\text{u},-}, \boldsymbol{\sigma}_t^{\text{s},+} = (\mathbf{1}_m - \mathbf{q}_t^{\text{u}}) \odot \boldsymbol{\sigma}_t^{\text{s},-} \text{ and } \boldsymbol{\sigma}_t^{\text{l},+} = \boldsymbol{\sigma}_t^{\text{l},-} - \mathbf{q}_t^{\text{l}} \odot \boldsymbol{\sigma}_t^{\text{s},-}, \tag{4}$$

where $\mathbf{1}_m$ denotes the $m$ dimensional vector consisting of ones. Again, we refer to the supplementary material for a more detailed derivations.

Besides avoiding the matrix inversion in the Kalman gain computation, the factorization of the covariance matrices reduces the total amount of numbers to store per matrix from $n^2$ to $3m$. Additionally, working with $\boldsymbol{\sigma}^{\text{s}}$ makes it trivial to ensure that the symmetry and positive definiteness of the state covariance are not affected by numerical issue.

### 2.5 LOSS AND TRAINING

We consider two different potential output distributions, Gaussian distributions for estimating low dimensional observations and Bernoulli distributions for predicting high dimensional observations/images. Both distributions are computed by decoders that use the current latent state estimate $\mathbf{z}_t$ as well its uncertainty estimates $\boldsymbol{\sigma}_t^{\text{u},+}, \boldsymbol{\sigma}_t^{\text{s},+}, \boldsymbol{\sigma}_t^{\text{l},+}$.

**Inferring states.** Let $\mathbf{s}_{1:T}$ be the ground truth sequence with dimension $D_s$, the Gaussian log-likelihood for a single sequence is then computed as

$$\mathcal{L}\left(\mathbf{s}_{(1:T)}\right) = \frac{1}{T} \sum_{t=1}^{T} \log \mathcal{N}\left(\mathbf{s}_t \middle| \text{dec}_\mu(\mathbf{z}_t^+), \text{dec}_\Sigma(\boldsymbol{\sigma}_t^{\text{u},+}, \boldsymbol{\sigma}_t^{\text{s},+}, \boldsymbol{\sigma}_t^{\text{l},+})\right), \tag{5}$$

where $\text{dec}_\mu(\cdot)$ and $\text{dec}_\Sigma(\cdot)$ denote the parts of the decoder that are responsible for decoding the latent mean and latent variance respectively.

**Inferring images.** Let $\mathbf{o}_{1:T}$ be images with $D_{\text{o}}$ pixels. The Bernoulli log-likelihood for a single sequence is then given by

$$\mathcal{L}\left(\mathbf{o}_{(1:T)}\right) = \frac{1}{T} \sum_{t=1}^{T} \sum_{i=0}^{D_{\text{o}}} o_t^{(i)} \log\left(\text{dec}_{o,i}\left(\mathbf{z}_t^+\right)\right) + \left(1 - o_t^{(i)}\right) \log\left(1 - \text{dec}_{o,i}\left(\mathbf{z}_t^+\right)\right), \tag{6}$$

where $o_t^{(i)}$ is the $i$th pixel of the $t$th image. The pixels are in this case represented by grayscale values in the range of $[0; 1]$. The $i$th dimension of the decoder is denoted by $\mathrm{dec}_{o,i}\left(\mathbf{z}_t^+\right)$, where we use a sigmoid transfer function as output units for the decoder.

Gradients are computed using (truncated) backpropagation through time (BPTT) (Werbos, 1990) and clipped. We optimize the objective using the Adam (Kingma and Ba, 2014) stochastic gradient descent optimizer with default parameters.

## 2.6 THE RECURRENT KALMAN NETWORK

The prediction and observation updates results in a new type of recurrent neural network, that we call *Recurrent Kalman Network*, which allows working in high dimensional state spaces while keeping numerical stability, computational efficiency and (relatively) low memory consumption. Similar to the input gate in LSTMs (Hochreiter and Schmidhuber, 1997) and GRUs (Cho et al., 2014) the Kalman gain can be seen as a gate controlling how much the current observation influences the state. However, this gating explicitly depends on the uncertainty estimates of the latent state and observation and is computed in a principled manner. Using sparse transition models allows working in higher dimensional spaces with considerably less parameters than LSTMs or GRUs. For a fixed bandwidth $b$ and number of basis matrices $k$, the number of parameters of the RKN scales linear in the state size while it scales quadratically for LSTMs and GRUs. Moreover, the RKN provides a principled method to deal with absent inputs by just omitting the update step and setting the posterior to the prior.

## 3 EVALUATION AND EXPERIMENTS

A full listing of hyperparameters and data set specifications can be found in the supplementary material[1]. We compare to LSTM and GRU baselines for which we replaced the RKN transition layer with generic LSTM and GRU layers. Those were given the encoder output as inputs and have an internal state size of $2n$. The internal state was split into two equally large parts, the first part was used to compute the mean and the second to compute the variance. We additionally executed most of the following experiments using the root mean squared error to illustrate that our approach is also competitive in prediction accuracy. The RMSE results can be found in the appendix.

### 3.1 PENDULUM

We evaluated the RKN on a simple simulated pendulum with images of size $24 \times 24$ pixels as observations. Gaussian transition noise with standard deviation of $\sigma = 0.1$ was added to the angular velocity after each step. In the first experiment, we evaluated filtering in the presence of high observation noise. We compare against LSTMs and GRUs as these are the only methods that can also perform state estimation. The amount of noise varies between no noise at all and the whole image consisting of pure noise. Furthermore, the noise is correlated over time, i.e., the model may observe pure noise for several consecutive time steps. Details about the noise sampling process can be found in the appendix. We represent the joint angle $\theta_t$ as a two dimensional vector $\mathbf{s}_t = \boldsymbol{\theta}_t = \left(\sin(\theta_t), \cos(\theta_t)\right)^T$ to avoid discontinuities. We compared different instances of our model to evaluate the effects of assuming sparse transition models and factorized state covariances. The results are given in Table 2. The results show that our assumptions do not affect the performance, while we need to learn less parameters and can use much more efficient computations using the factorized representation. Note that for the more complex experiments with a higher dimensional latent state space, we were not able to experiment with full covariance matrices due to lack of memory and massive computation times. Moreover, the RKN outperforms LSTM and GRU in all settings.

In a second experiment we evaluated the image prediction performance and compare against existing variational inference approaches. We randomly removed half of the images from the sequences and tasked the models with imputing those missing frames, i.e., we train the models to predict images instead of the position. We compared our approach to the Kalman Variational Autoencoder

---

[1] A link to source code will be added here in the final version.

| Model | Log Likelihood | Model | Log Likelihood |
|---|---|---|---|
| RKN ($m = 15, b = 3, K = 15$) | $6.182 \pm 0.155$ | LSTM $m = 50$ | $5.773 \pm 0.231$ |
| RKN $m = b = 15, K = 3$) | $6.248 \pm 0.1715$ | LSTM $m = 6$ | $6.019 \pm 0.122$ |
| RKN ($m = 15, b = 3, K = 15$, fc ) | $6.161 \pm 0.23$ | GRU $m = 50$ | $5.649 \pm 0.197$ |
| RKN ($m = b = 15, K = 15$, fc ) | $6.197 \pm 0.249$ | GRU $m = 8$ | $6.051 \pm 0.145$ |

Table 2: Our approach outperforms the generic LSTM and GRU Baselines. The GRU with $m = 8$ and the LSTM with $m = 6$ where designed to have roughly the same amount of parameters as the RKN with $b = 3$. In the case where $m = b$ the RKN uses a full transition matrix. *fc* stands for full covariance matrix, i.e., we do not use factorization of the belief state.

| Model | Log Likelihood | Model | Log Likelihood |
|---|---|---|---|
| RKN (informed) | $-12.782 \pm 0.0160$ | KVAE (informed, filter) | $-14.383 \pm 0.229$ |
| RKN (uninformed) | $-12.788 \pm 0.0142$ | KVAE (informed, smooth) | $-13.337 \pm 0.236$ |
| E2C | $-95.539 \pm 1.754$ | KVAE (uninformed, filter) | $-46.320 \pm 6.488$ |
| SIN | $-101.268 \pm 0.567$ | KVAE (uninformed, smooth) | $-38.170 \pm 5.399$ |

Table 3: Comparison on the image imputation task. The informed models where given a mask of booleans indicating which images are valid and which not. The uninformed models where given a black image whenever the image was not valid. E2C and SIN only work in the informed case. Since the KVAE is capable of smoothing in addition to filtering, we evaluated both. Our approach outperforms all models. Only the informed KVAE yields comparable, but still slightly worse results while E2C and SIN fail to capture the dynamics. The uninformed KVAE fails at identifying the non valid images.

(KVAE) (Fraccaro et al., 2017), Embed to Control (E2C) (Watter et al., 2015) and Structured Inference Networks (SIN) (Krishnan et al., 2017). The results can be found in Table 3. Again, our RKN outperforms all other models. This is surprising as the variational inference models use much more complex inference methods and in some cases even more information such as in the KVAE smoothing case. Sample sequences can be found in the supplementary material. All hyperparameters are the same as for the previous experiment.

## 3.2 MULTIPLE PENDULUMS

Dealing with uncertainty in a principled manner becomes even more important if the observation noise affects different parts of the observation in different ways. In such a scenario the model has to infer which parts are currently observable and which parts are not. To obtain a simple experiment with that property, we repeated the pendulum experiments with three colored pendulums in one image. The image was split into four equally sized square parts and the noise was generated individually for each part such that some pendulums may be occluded while others are visible. Exemplary images are shown in the supplementary material. A comparison to LSTM and GRU baselines can be found in Figure 2 and an exemplary trajectory in Figure 3. The RKN again clearly outperforms the competing methods. We also computed the quality of the uncertainty prediction by showing the histograms of the normalized prediction errors. While this histogram has clearly a Gaussian shape for the RKN, it looks like a less regular distribution for the LSTMs.

## 3.3 QUAD LINK

We repeated the filtering experiment on a system with much more complicated dynamics, a quad link pendulum on images of size $48 \times 48$ pixels. Since the individual links of the quad link may occlude each other different amounts of noise are induced for each link. Two versions of this experiment were evaluated. One without additional noise and one were we added noise generated by the same process used in the pendulum experiments. You can find the results in Table 4.

Furthermore, we repeated the imputation experiment with the quad link. We compared only to the informed KVAE, since it was the only model producing competitive results for the pendulum. Our approach achieved $-44.470 \pm 0.208$ (informed) and $-44.584 \pm 0.236$ (uninformed). The KVAE

| Model | Log Likelihood |
|---|---|
| RKN $m = 45$ $b = 3, k = 15$ | $11.51 \pm 1.703$ |
| LSTM $m = 50$ | $7.5224 \pm 1.564$ |
| LSTM $m = 12$ | $7.429 \pm 1.307$ |
| GRU $m = 50$ | $7.541 \pm 1.547$ |
| GRU $m = 12$ | $5.602 \pm 1.468$ |

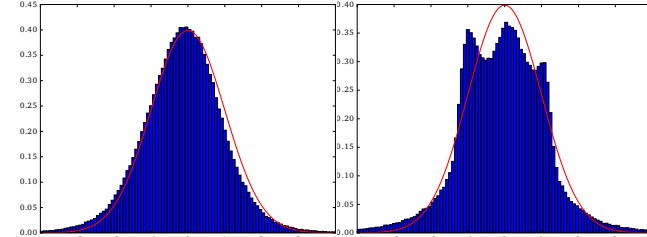

Figure 2: Results of the multiple pendulum experiments. To evaluate the quality of our uncertainty prediction we compute the normalized error $\frac{s^{(j)} - s^{(j),+}}{\sigma^{(j),+}}$ for each entry $j$ of $\mathbf{s}$ for all time steps in all test sequences. This normalized error should follow a Gaussian distribution with unit variance if the prediction is correct. We compare the resulting error histograms with the a unit variance Gaussian. The left histogram shows the RKN, the right one the LSTM. The RKN perfectly fits the normal distribution while the LSTM's normalized error distribution has several modes. Again we designed the smaller LSTM and GRU to have roughly the same amount of parameters as the RKN.

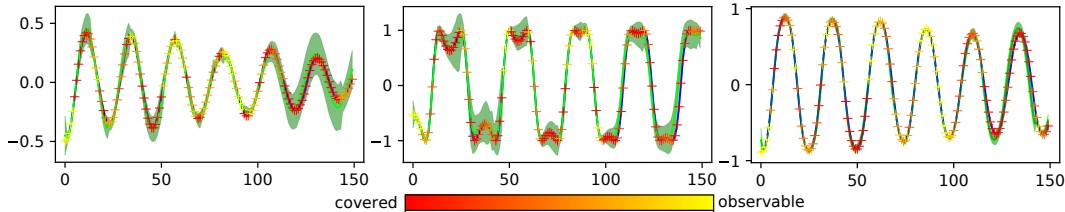

Figure 3: Predicted sine value of the tree links with 2 times standard deviation (green). Ground truth displayed in blue. The crosses visualize the current visibility of the link with yellow corresponding to fully visible and red to fully occluded. If there is no observation for a considerable time the predictions become less precise due to transition noise, however the variance also increases.

achieved $-52.608 \pm 0.602$ for smoothing and $-59.0218 \pm 0.580$ for filtering . Sample images can be found in the appendix.

## 3.4   KITTI DATASET FOR VISUAL ODOMETRY

We evaluated the RKN on the KITTI Data set for visual odometry (Geiger et al., 2012). Following (Zhao et al., 2018) we constructed an encoder consisting of FlowNet2 (Ilg et al., 2017), four convolutional and two branches of fully connected layers.

Following the standard KITTI evaluation procedure achieve an transnational error of $4.24\%$ and a rotational error of $0.0404$rad /100m . Those result are competitive to other recent results on the KITTI dataset using only monocular images (Zhao et al., 2018; Wang et al., 2018) and demonstrate the applicability of our approach to real world applications.

| Model | without noise Log Likelihood | with noise Log Likelihood |
|---|---|---|
| RKN $(m = 100, b = 3, K = 15)$ | $14.534 \pm 0.176$ | $6.259 \pm 0.412$ |
| LSTM $(m = 50)$ | $11.960 \pm 1.24$ | $5.21 \pm 0.305$ |
| LSTM $(m = 100)$ | $7.858 \pm 4.680$ | $3.87 \pm 0.938$ |
| GRU $(m = 50)$ | $10.346 \pm 2.70$ | $4.696 \pm 0.699$ |
| GRU $(m = 100)$ | $5.82 \pm 2.80$ | $1.2 \pm 1.105$ |

Table 4: Comparison of our approach with the LSTM and GRU Baselines on the Quad Link Pendulum. Again the RKN performs significantly better than LSTM and GRU who fail to perform well.

## 4 CONCLUSION

In this paper, we introduced the *Recurrent Kalman Network* that jointly learns high-dimensional representations of the system in a latent space with locally linear transition models and factorized covariances. The update equations in the high-dimensional space are based on the update equations of the classical Kalman filter, however due to the factorization assumptions they simplify to scalar operations that can be performed much faster and with greater numerical stability. Our model outperforms generic LSTMs and GRUs on various state estimation tasks while providing reasonable uncertainty estimates. Additionally, it outperformed several generative models on an image imputation task. Training is straight forward and can be done in an end-to-end fashion.

In future work we want to exploit the principled notion of a variance provided by our approach in scenarios where such a notion is beneficial, e.g. reinforcement learning. Similar to (Fraccaro et al., 2017) we could expand our approach to not just filter but smooth over trajectories in offline post-processing scenarios which could potentially increase the estimation performance significantly.

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

APPENDIX

The Kalman filter (Kalman et al., 1960) works by iteratively applying two steps, predict and update. It assumes additive Gaussian noise with zero mean and covariances $\mathbf{\Sigma}^{\text{trans}}$ and $\mathbf{\Sigma}^{\text{obs}}$ on both transitions and observations, which need to be given to the filter. During the prediction step the transition model $\mathbf{A}$ is used to infer the next prior state estimate $\left(\mathbf{x}_t^-, \mathbf{\Sigma}_t^-\right)$, i.e., a-priori to the observation, from the last posterior estimate $\left(\mathbf{x}_{t-1}^+, \mathbf{\Sigma}_{t-1}^+\right)$, by

$$\mathbf{x}_t^- = \mathbf{A}\mathbf{x}_{t-1}^+ \quad \text{and} \quad \mathbf{\Sigma}_t^- = \mathbf{A}\mathbf{\Sigma}_{t-1}^+\mathbf{A}^T + \mathbf{\Sigma}^{\text{trans}}.$$

The prior estimate is then updated using the current observation $\mathbf{w}_t$ and the observation model $\mathbf{H}$ to obtain the posterior estimate $\left(\mathbf{x}_t^+, \mathbf{\Sigma}_t^+\right)$, i.e.,

$$\mathbf{x}_t^+ = \mathbf{x}_t^- + \mathbf{Q}_t\left(\mathbf{w}_t - \mathbf{H}\mathbf{x}_t^-\right) \text{ and } \mathbf{\Sigma}_t^+ = (\mathbf{I} - \mathbf{Q}_t\mathbf{H})\,\mathbf{\Sigma}_t^-, \text{ with } \mathbf{Q}_t = \mathbf{\Sigma}_t^-\mathbf{H}^T\left(\mathbf{H}\mathbf{\Sigma}_t^-\mathbf{H}^T + \mathbf{\Sigma}^{\text{obs}}\right)^{-1},$$

where $\mathbf{I}$ denotes the identity matrix. The matrix $\mathbf{Q}_t$ is referred to as the Kalman gain. The whole update step can be interpreted as a weighted average between state and observation estimate, where the weighting, i.e., $\mathbf{Q}_t$, depends on the uncertainty about those estimates. If currently no observation is present or future states should be predicted, the update step is omitted.

## A  SIMPLIFIED KALMAN FILTER FORMULAS

As stated above the simple latent observation model $\mathbf{H} = \left[\begin{array}{cc} \mathbf{I}_m & \mathbf{0}_{m\times(n-m)} \end{array}\right]$, as well as the assumed factorization of the covariance matrices allow us to simplify the Kalman Filter equations.

### A.1  NOTATION

In the following derivations we neglect the time indices $t$ and $t+1$ for brevity. For any matrix $\mathbf{M}$, $\hat{\mathbf{M}}$ denotes a diagonal matrix with the same diagonal as $\mathbf{M}$, $\mathbf{m}$ denotes a vector containing those diagonal elements and $M^{(ij)}$ denotes the entry at row $i$ and column $j$. Similarly, $v^{(i)}$ denotes the $i$-th entry of a vector $\mathbf{v}$. The point wise product between two vectors of same length (Hadamat Product) will be denoted by $\odot$ and the point wise division by $\oslash$.

### A.2  PREDICTION STEP

$$\text{Mean:} \qquad\qquad \mathbf{z}^- = \mathbf{A}\mathbf{z}^+$$
$$\text{Covariance:} \qquad\qquad \mathbf{\Sigma}^- = \mathbf{A}\mathbf{\Sigma}^+\mathbf{A}^T + \hat{\mathbf{\Sigma}}^{\text{trans}}$$

The computation of the mean can not be further simplified, however, depending on the state size and bandwidth, sparse matrix multiplications may be exploited. For the covariance, let $\mathbf{T} = \mathbf{A}\mathbf{\Sigma}^+$. Then,

$$\mathbf{T} = \mathbf{A}\mathbf{\Sigma}^+ = \left[\begin{array}{cc} \mathbf{B}_{11} & \mathbf{B}_{12} \\ \mathbf{B}_{21} & \mathbf{B}_{22} \end{array}\right]\left[\begin{array}{cc} \hat{\mathbf{\Sigma}}^{u,+} & \hat{\mathbf{\Sigma}}^{s,+} \\ \hat{\mathbf{\Sigma}}^{s,+} & \hat{\mathbf{\Sigma}}^{l,+} \end{array}\right]$$

$$= \left[\begin{array}{cc} \mathbf{B}_{11}\hat{\mathbf{\Sigma}}^{u,+} + \mathbf{B}_{12}\hat{\mathbf{\Sigma}}^{s,+} & \mathbf{B}_{11}\hat{\mathbf{\Sigma}}^{s,+} + \mathbf{B}_{12}\hat{\mathbf{\Sigma}}^{l,+} \\ \mathbf{B}_{21}\hat{\mathbf{\Sigma}}^{u,+} + \mathbf{B}_{22}\hat{\mathbf{\Sigma}}^{s,+} & \mathbf{B}_{21}\hat{\mathbf{\Sigma}}^{s,+} + \mathbf{B}_{22}\hat{\mathbf{\Sigma}}^{l,+} \end{array}\right]$$

and

$$\mathbf{\Sigma}^- = \mathbf{T}\mathbf{A}^T + \hat{\mathbf{\Sigma}}^{\text{trans}}$$

$$= \left[\begin{array}{cc} \mathbf{B}_{11}\hat{\mathbf{\Sigma}}^{u,+} + \mathbf{B}_{12}\hat{\mathbf{\Sigma}}^{s,+} & \mathbf{B}_{11}\hat{\mathbf{\Sigma}}^{s,+} + \mathbf{B}_{12}\hat{\mathbf{\Sigma}}^{l,+} \\ \mathbf{B}_{21}\hat{\mathbf{\Sigma}}^{u,+} + \mathbf{B}_{22}\hat{\mathbf{\Sigma}}^{s,+} & \mathbf{B}_{21}\hat{\mathbf{\Sigma}}^{s,+} + \mathbf{B}_{22}\hat{\mathbf{\Sigma}}^{l,+} \end{array}\right]\left[\begin{array}{cc} \mathbf{B}_{11}^T & \mathbf{B}_{21}^T \\ \mathbf{B}_{12}^T & \mathbf{B}_{22}^T \end{array}\right] + \hat{\mathbf{\Sigma}}^{\text{trans}}$$

$$= \left[\begin{array}{cc} \hat{\mathbf{\Sigma}}^{u,-} & \hat{\mathbf{\Sigma}}^{s,-} \\ \hat{\mathbf{\Sigma}}^{s,-} & \hat{\mathbf{\Sigma}}^{l,-} \end{array}\right],$$

with

$$\boldsymbol{\Sigma}^{\mathrm{u},-} = \left(\mathbf{B}_{11}\hat{\boldsymbol{\Sigma}}^{\mathrm{u},+} + \mathbf{B}_{12}\hat{\boldsymbol{\Sigma}}^{\mathrm{s},+}\right)\mathbf{B}_{11}^T + \left(\mathbf{B}_{11}\hat{\boldsymbol{\Sigma}}^{\mathrm{s},+} + \mathbf{B}_{12}\hat{\boldsymbol{\Sigma}}^{\mathrm{l},+}\right)\mathbf{B}_{12}^T + \hat{\boldsymbol{\Sigma}}^{\mathrm{u,trans}}$$

$$= \mathbf{B}_{11}\hat{\boldsymbol{\Sigma}}^{\mathrm{u},+}\mathbf{B}_{11}^T + \mathbf{B}_{12}\hat{\boldsymbol{\Sigma}}^{\mathrm{s},+}\mathbf{B}_{11}^T + \mathbf{B}_{11}\hat{\boldsymbol{\Sigma}}^{\mathrm{s},+}\mathbf{B}_{12}^T + \mathbf{B}_{12}\hat{\boldsymbol{\Sigma}}^{\mathrm{l},+}\mathbf{B}_{12}^T + \hat{\boldsymbol{\Sigma}}^{\mathrm{u,trans}}$$

$$\boldsymbol{\Sigma}^{\mathrm{l},-} = \left(\mathbf{B}_{21}\hat{\boldsymbol{\Sigma}}^{\mathrm{u},+} + \mathbf{B}_{22}\hat{\boldsymbol{\Sigma}}^{\mathrm{s},+}\right)\mathbf{B}_{21}^T + \left(\mathbf{B}_{21}\hat{\boldsymbol{\Sigma}}^{\mathrm{s},+} + \mathbf{B}_{22}\hat{\boldsymbol{\Sigma}}^{\mathrm{l},+}\right)\mathbf{B}_{22}^T + \hat{\boldsymbol{\Sigma}}^{\mathrm{l,trans}}$$

$$= \mathbf{B}_{21}\hat{\boldsymbol{\Sigma}}^{\mathrm{u},+}\mathbf{B}_{21}^T + \mathbf{B}_{22}\hat{\boldsymbol{\Sigma}}^{\mathrm{s},+}\mathbf{B}_{21}^T + \mathbf{B}_{21}\hat{\boldsymbol{\Sigma}}^{\mathrm{s},+}\mathbf{B}_{22}^T + \mathbf{B}_{22}\hat{\boldsymbol{\Sigma}}^{\mathrm{l},+}\mathbf{B}_{22}^T + \hat{\boldsymbol{\Sigma}}^{\mathrm{l,trans}}$$

$$\boldsymbol{\Sigma}^{\mathrm{s},-} = \left(\mathbf{B}_{21}\hat{\boldsymbol{\Sigma}}^{\mathrm{u},+} + \mathbf{B}_{22}\hat{\boldsymbol{\Sigma}}^{\mathrm{s},+}\right)\mathbf{B}_{11}^T + \left(\mathbf{B}_{21}\hat{\boldsymbol{\Sigma}}^{\mathrm{s},+} + \mathbf{B}_{22}\hat{\boldsymbol{\Sigma}}^{\mathrm{l},+}\right)\mathbf{B}_{12}^T$$

$$= \mathbf{B}_{21}\hat{\boldsymbol{\Sigma}}^{\mathrm{u},+}\mathbf{B}_{11}^T + \mathbf{B}_{22}\hat{\boldsymbol{\Sigma}}^{\mathrm{s},+}\mathbf{B}_{11}^T + \mathbf{B}_{21}\hat{\boldsymbol{\Sigma}}^{\mathrm{s},+}\mathbf{B}_{12}^T + \mathbf{B}_{22}\hat{\boldsymbol{\Sigma}}^{\mathrm{l},+}\mathbf{B}_{12}^T$$

Since we are only interested in the diagonal parts of $\boldsymbol{\Sigma}^{\mathrm{u},-}$, $\boldsymbol{\Sigma}^{\mathrm{l},-}$ and $\boldsymbol{\Sigma}^{\mathrm{s},-}$ i.e. $\hat{\boldsymbol{\Sigma}}^{\mathrm{u},-}$, $\hat{\boldsymbol{\Sigma}}^{\mathrm{l},-}$ and $\hat{\boldsymbol{\Sigma}}^{\mathrm{s},-}$, we can further simplify these equations by realizing two properties of the terms above. First, for any matrix $\mathbf{M}$, $\mathbf{N}$ and a diagonal matrix $\hat{\boldsymbol{\Sigma}}$ it holds that

$$\left(\mathbf{M}\hat{\boldsymbol{\Sigma}}\mathbf{N}^T\right)^{(ii)} = \sum_{k=1}^{n} A^{(ik)}B^{(ik)}\sigma^{(k)} = \left(\mathbf{N}\hat{\boldsymbol{\Sigma}}\mathbf{M}^T\right)_{ii}.$$

Hence, we can simplify the equations for the upper and lower part to

$$\boldsymbol{\Sigma}^{u,-} = \mathbf{B}_{11}\hat{\boldsymbol{\Sigma}}^{u,+}\mathbf{B}_{11}^T + 2\cdot\mathbf{B}_{12}\hat{\boldsymbol{\Sigma}}^{s,+}\mathbf{B}_{11}^T + \mathbf{B}_{12}\hat{\boldsymbol{\Sigma}}^{l,+}\mathbf{B}_{12}^T + \hat{\boldsymbol{\Sigma}}^{u,trans},$$

$$\boldsymbol{\Sigma}^{l,-} = \mathbf{B}_{21}\hat{\boldsymbol{\Sigma}}^{u,+}\mathbf{B}_{21}^T + 2\cdot\mathbf{B}_{22}\hat{\boldsymbol{\Sigma}}^{s,+}\mathbf{B}_{21}^T + \mathbf{B}_{22}\hat{\boldsymbol{\Sigma}}^{l,+}\mathbf{B}_{22}^T + \hat{\boldsymbol{\Sigma}}^{l,trans}.$$

Second, since we are only interested in the diagonal of the result it is sufficient to compute only the diagonals of the individual parts of the sums which are almost all of the same structure i.e. $\mathbf{S} = \mathbf{M}\hat{\boldsymbol{\Sigma}}^+\mathbf{N}^T$. Let $\mathbf{T} = \mathbf{M}\hat{\boldsymbol{\Sigma}}^+$, then each element of $\mathbf{T}$ can be computed as

$$T^{(ij)} = \sum_{k=1}^{n} M^{(ik)}\hat{\Sigma}^{(kj)} = M^{(ij)}\sigma^{(j)}.$$

Consequently, the elements of $\mathbf{S} = \mathbf{T}\mathbf{A}^T$ can be computed as

$$S^{(ij)} = \sum_{k=1}^{n} T^{(ik)}A^{(kj)} = \sum_{k=1}^{n} M^{(ik)}\sigma_k N^{(jk)}.$$

Ultimately, we are not interested in $\mathbf{S}$ but only in $\hat{\mathbf{S}}$

$$\hat{S}^{(ii)} = \sum_{k=1}^{n} M^{(ik)}N^{(ik)}\sigma^{(}k).$$

Using this we obtain can obtain the entries of $\boldsymbol{\sigma}^{u,-}$, $\boldsymbol{\sigma}^{l,-}$ and $\boldsymbol{\sigma}^{s,-}$ by

$$\sigma^{u,-,(i)} = \sum_{k=1}^{m}\left(B_{11}^{(ik)}\right)^2\sigma^{u,+,(i)} + 2\sum_{k=1}^{m}B_{11}^{(ik)}B_{12}^{(ik)}\sigma^{s,+,(i)} + \sum_{k=1}^{m}\left(B_{12}^{(ik)}\right)^2\sigma^{l,+,(i)} + \sigma^{u,\mathrm{trans},(i)}$$

$$\tag{7}$$

$$\sigma^{l,-,(i)} = \sum_{k=1}^{m}\left(B_{21}^{(ik)}\right)\sigma^{u,+,(i)} + 2\sum_{k=1}^{m}B_{22}^{(ik)}B_{21}^{(ik)}\sigma^{s,+,(i)} + \sum_{k=1}^{m}\left(B_{22}^{(ik)}\right)^2\sigma^{l,+,(i)} + \sigma^{l,\mathrm{trans},(i)}$$

$$\tag{8}$$

$$\sigma^{s,-,(i)} = \sum_{k=1}^{m}B_{21}^{(ik)}B_{11}^{(ik)}\sigma^{u,+,(i)} + \sum_{k=1}^{m}B_{22}^{(ik)}B_{11}^{(ik)}\sigma^{s,+,(i)}... \tag{9}$$

$$... + \sum_{k=1}^{m}B_{21}^{(ik)}B_{12}^{(ik)}\sigma^{s,+,(i)} + \sum_{k=1}^{m}B_{22}^{(ik)}B_{12}^{(ik)}\sigma^{l,+,(i)},$$

which can be implemented efficiently using elementwise matrix multiplication and sum reduction.

## A.3 UPDATE STEP

$$
\begin{aligned}
\text{Kalman Gain} \qquad & \mathbf{Q} = \boldsymbol{\Sigma}^{-}\mathbf{H}^{T}\left(\mathbf{H}\boldsymbol{\Sigma}^{-}\mathbf{H}^{T} + \boldsymbol{\Sigma}^{\mathrm{obs}}\right)^{-1} \\
\text{Mean} \qquad & \mathbf{z}^{+} = \mathbf{z}^{-} + \mathbf{Q}\left(\mathbf{w} - \mathbf{H}\mathbf{z}^{-}\right) \\
\text{Covariance} \qquad & \boldsymbol{\Sigma}^{+} = \left(\mathbf{I} - \mathbf{Q}\mathbf{H}\right)\boldsymbol{\Sigma}^{-}
\end{aligned}
$$

First, note that

$$
\boldsymbol{\Sigma}^{-}\mathbf{H}^{T} = \begin{bmatrix} \hat{\boldsymbol{\Sigma}}^{\mathrm{u},-} \\ \hat{\boldsymbol{\Sigma}}^{\mathrm{s},-} \end{bmatrix} \quad \text{and} \quad \mathbf{H}\boldsymbol{\Sigma}^{-}\mathbf{H}^{T} + \hat{\boldsymbol{\Sigma}}^{\mathrm{obs}} = \hat{\boldsymbol{\Sigma}}^{\mathrm{u},-} + \hat{\boldsymbol{\Sigma}}^{\mathrm{obs}}
$$

and thus the computation of the Kalman Gain only involves diagonal matrices. Hence the Kalman Gain matrix also consists of two diagonal matrices, i.e., $\mathbf{Q} = \begin{bmatrix} \hat{\mathbf{Q}}^{\mathrm{u}} \\ \hat{\mathbf{Q}}^{\mathrm{l}} \end{bmatrix}$ whose diagonals can be computed by

$$
\mathbf{q}^{\mathrm{u}} = \boldsymbol{\sigma}^{\mathrm{u},-} \oslash \left(\boldsymbol{\sigma}^{\mathrm{u},-} + \boldsymbol{\sigma}^{\mathrm{obs}}\right) \text{ and } \mathbf{q}^{\mathrm{l}} = \boldsymbol{\sigma}^{\mathrm{s},-} \oslash \left(\boldsymbol{\sigma}^{\mathrm{u},-} + \boldsymbol{\sigma}^{\mathrm{obs}}\right). \tag{10}
$$

Using this result, the mean update can be simplified to

$$
\mathbf{z}^{+} = \mathbf{z}^{-} + \begin{bmatrix} \mathbf{q}_{\mathrm{u}} \\ \mathbf{q}_{\mathrm{l}} \end{bmatrix} \odot \begin{bmatrix} \mathbf{w} - \mathbf{z}^{\mathrm{u},-} \\ \mathbf{w} - \mathbf{z}^{\mathrm{u},-} \end{bmatrix}. \tag{11}
$$

For the covariance we get:

$$
\begin{aligned}
\begin{bmatrix} \hat{\boldsymbol{\Sigma}}^{\mathrm{u},+} & \hat{\boldsymbol{\Sigma}}^{\mathrm{s},+} \\ \hat{\boldsymbol{\Sigma}}^{\mathrm{s},+} & \hat{\boldsymbol{\Sigma}}^{\mathrm{l},+} \end{bmatrix} &= \left(\mathbf{I}_{n} - \mathbf{Q}\mathbf{H}\right)\boldsymbol{\Sigma}^{-} = \begin{bmatrix} \mathbf{I}_{m} - \hat{\mathbf{Q}}^{\mathrm{u}} & \mathbf{0}_{m \times m} \\ -\hat{\mathbf{Q}}^{\mathrm{l}} & \mathbf{I}_{m} \end{bmatrix}\begin{bmatrix} \hat{\boldsymbol{\Sigma}}^{\mathrm{u},-} & \hat{\boldsymbol{\Sigma}}^{\mathrm{s},-} \\ \hat{\boldsymbol{\Sigma}}^{\mathrm{s},-} & \hat{\boldsymbol{\Sigma}}^{\mathrm{l},-} \end{bmatrix} \\
&= \begin{bmatrix} \left(\mathbf{I}_{m} - \hat{\mathbf{Q}}^{\mathrm{u}}\right)\hat{\boldsymbol{\Sigma}}^{\mathrm{u},-} & \left(\mathbf{I}_{m} - \hat{\mathbf{Q}}^{\mathrm{u}}\right)\hat{\boldsymbol{\Sigma}}^{\mathrm{s},-} \\ -\hat{\mathbf{Q}}^{\mathrm{l}}\hat{\boldsymbol{\Sigma}}^{\mathrm{u},-} + \hat{\boldsymbol{\Sigma}}^{\mathrm{s},-} & -\hat{\mathbf{Q}}^{\mathrm{l}}\hat{\boldsymbol{\Sigma}}^{\mathrm{s},-} + \hat{\boldsymbol{\Sigma}}^{\mathrm{l},-} \end{bmatrix}.
\end{aligned}
$$

Hence the diagonals of the individual parts can be computed as

$$
\boldsymbol{\sigma}^{\mathrm{u},+} = \left(\mathbf{1}_{m} - \mathbf{q}^{\mathrm{u}}\right) \odot \boldsymbol{\sigma}^{\mathrm{u},-} \tag{12}
$$

$$
\boldsymbol{\sigma}^{\mathrm{s},+} = \left(\mathbf{1}_{m} - \mathbf{q}^{\mathrm{u}}\right) \odot \boldsymbol{\sigma}^{\mathrm{s},-} \tag{13}
$$

$$
\boldsymbol{\sigma}^{\mathrm{l},+} = \boldsymbol{\sigma}^{\mathrm{l},-} - \mathbf{q}^{\mathrm{l}} \odot \boldsymbol{\sigma}^{\mathrm{s},-}. \tag{14}
$$

## B ROOT MEAN SQUARE ERROR RESULTS

To evaluate the actual prediction performance of our approach we repeated some experiments using the RMSE as loss function. Other than that and removing the variance output of the decoder no changes were made to the model, hyperparameters and learning procedure. The results can be found in Table 5.

## C VISUALIZATION OF IMPUTATION RESULTS

Exemplary results of the data imputation experiments conducted for the Pendulum and Quad Link experiment can be found in Figure 4

## D NETWORK ARCHITECTURES AND HYPER PARAMETERS

For all experiments Adam (Kingma and Ba, 2014) with default parameters ($\alpha = 10^{-3}$, $\beta_{1} = 0.9$, $\beta_{2} = 0.999$ and $\varepsilon = 10^{-8}$) was used as an optimizer. The gradients were computed using (truncated) Backpropagation Trough Time (BpTT) (Werbos, 1990). Further in all (transposed) convolutional layers layer normalization (LN) (Ba et al., 2016) was employed to normalize the filter responses. "Same" padding was used. The elu activation function (Clevert et al., 2015) plus a constant 1 is denoted by (elu + 1) was used to ensure that the variance outputs are positive.

| Model | Root Mean Squared Error |
|---|---|
| Pendulum | |
| RKN ($m = 15, b = 3, K = 15$) | $0.0779 \pm 0.0082$ |
| RKN ($m = b = 15, K = 15$) | $0.0758 \pm 0.0094$ |
| LSTM ($m = 50$) | $0.0920 \pm 0.0774$ |
| LSTM ($m = 6$) | $0.0959 \pm 0.0100$ |
| GRU ($m = 50$) | $0.0821 \pm 0.0084$ |
| GRU ($m = 8$) | $0.0916 \pm 0.0087$ |
| Multiple Pendulums | |
| RKN ($m = 45, b = 3, k = 15$) | $0.0878 \pm 0.0036$ |
| LSTM ($m = 50$) | $0.098 \pm 0.0036$ |
| LSTM ($m = 12$) | $0.104 \pm 0.0043$ |
| GRU ($m = 50$) | $0.112 \pm 0.0371$ |
| GRU ($m = 14$) | $0.105 \pm 0.0055$ |
| Quad Link (without additional noise ) | |
| RKN ($m = 100, b = 25, k = 15$) | $0.103 \pm 0.00076$ |
| LSTM ($m = 100$) | $0.175 \pm 0.182$ |
| LSTM ($m = 25$) | $0.118 \pm 0.0049$ |
| GRU ($m = 100$) | $0.278 \pm 0.105$ |
| GRU ($m = 25$) | $0.121 \pm 0.0021$ |
| Quad Link (with additional noise ) | |
| RKN ($m = 100, b = 25, k = 15$) | $0.171 \pm 0.0039$ |
| LSTM ($m = 75$) | $0.175 \pm 0.0022$ |
| GRU ($m = 25$) | $0.204 \pm 0.0023$ |

Table 5: RMSE Results

### D.1 Pendulum and Multiple Pendulum Experiments

**Observations** *Pendulum*: Grayscale images of size $24 \times 24$ pixels. *Multiple Pendulum*: RGB images of size $24 \times 24$ pixels. See Figure 5 for examples.

**Dataset:** 1000 Train and 500 Test sequences of length 150. For the filtering experiments noise according to section E was added , for imputation 50% of the images were removed randomly.

**Encoder:** 2 convolution + 1 fully connected + linear output & (elu + 1) output:

- Convolution 1: 12, $5 \times 5$ filter, ReLU, $2 \times 2$ max pool with $2 \times 2$ stride
- Convolution 2: 12, $3 \times 3$ filter with $2 \times 2$ stride, ReLU, $2 \times 2$ max pool with $2 \times 2$ stride
- *Pendulum*: Fully Connected 1: 30, ReLU
- *Multiple Pendulum*: Fully Connected 1: 90, ReLU

**Transition Model** *Pendulum*: 15 dimensional latent observation, 30 dimensional latent state. *Multiple Pendulum*: 45 dimensional latent observation, 90 dimensonal latent state. *Both:* bandwidth: 3, number of basis: 15

- $\alpha(\mathbf{z}_t)$: No hidden layers - softmax output

**Decoder** (for $\mathbf{s}_t^+$): 1 fully connected + linear output:

- Fully Connected 1: 10, ReLU

**Decoder** (for $\mathbf{o}_t^+$): 1 fully connected + 2 transposed convolution + transposed convolution output:

- Fully Connected 1: 144 ReLU
- Transposed Convolution 1: 16, $5 \times 5$ filter with $4 \times 4$ stride, ReLU
- Transposed Convolution 2: 12, $3 \times 3$ filter with $2 \times 2$ stride, ReLU
- Transposed Convolution Out: *Pendulum*: 1 *Multiple Pendulum*: 3, $3 \times 3$ filter with $1 \times 1$ stride, Sigmoid

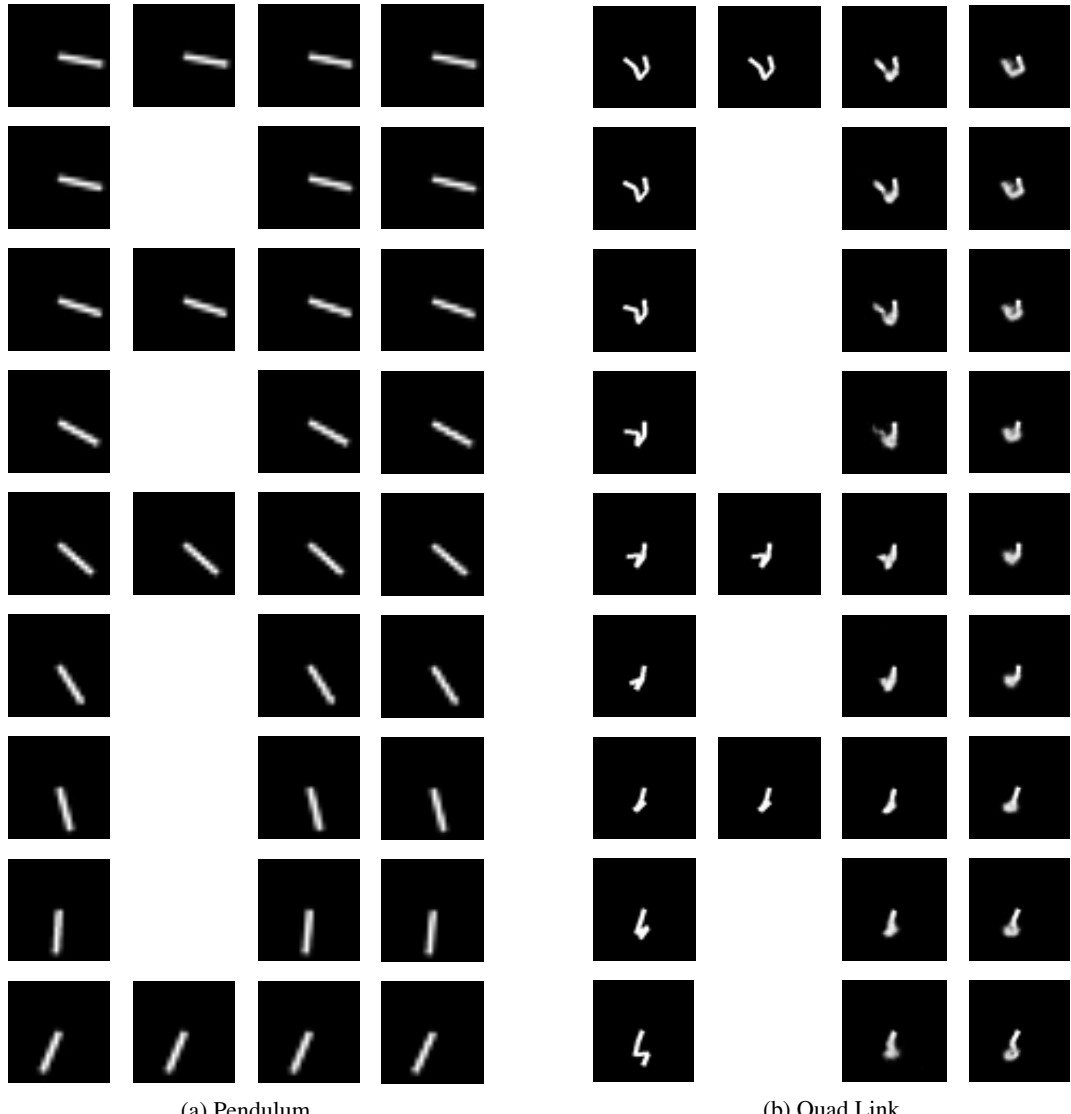

(a) Pendulum        (b) Quad Link

Figure 4: Each of (a) and (b) shows from left to right: true images, input to the models, imputation results for RKF, imputation results for KVAE(Fraccaro et al., 2017).

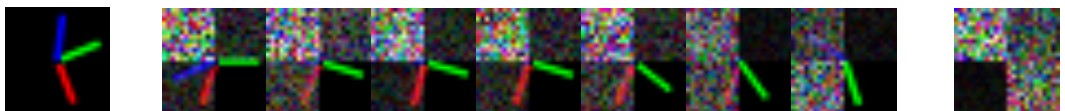

Figure 5: Example images for the multiple pendulum experiments. **Left**: Noise free image. **Middle**: sequence of images showing how the noise affects different pendulums differently. **Right**: Image without useful information.

**Decoder** (for $\boldsymbol{\sigma}_t^+$ or $\sigma_t^+$): 1 fully connected + (elu + 1):

- Fully Connected 1: 10, ReLU

### D.2 QUAD LINK

**Observations**: Grayscale images of size $48x48$ pixels.
**Dataset**: 4000 Train and 1000 Test sequences of length 150. For the filtering with additional noise experiments noise according to section D was added, for imputation $50\%$ of the images were removed randomly.
**Encoder:** 2 convolution + 1 fully connected + linear output & (elu + 1) output:

- Convolution 1: 12, $5 \times 5$ filter with $2 \times 2$ stride, ReLU, $2 \times 2$ max pool with $2 \times 2$ stride
- Convolution 2: 12, $3 \times 3$ filter with $2 \times 2$ stride, ReLU, $2 \times 2$ max pool with $2 \times 2$ stride
- Fully Connected 1: 200 ReLU

**Transition Model** 100 dimensional latent observation, 200 dimensional latent state, bandwidth: 3, number of basis: 15

- $\alpha(\mathbf{z}_t)$: No hidden layers - softmax output

**Decoder** (for $\mathbf{s}_t^+$): 1 fully connected + linear output:

- Fully Connected 1: 10, ReLU

**Decoder** (for $\mathbf{o}_t^+$): 1 fully connected + 2 transposed convolution + transposed convolution output:

- Fully Connected 1: 144 ReLU
- Transposed Convolution 1: 16, $5 \times 5$ filter with $4 \times 4$ stride, ReLU
- Transposed Convolution 2: 12, $3 \times 3$ filter with $4 \times 4$ stride, ReLU
- Transposed Convolution Out: 1, $1 \times 1$ stride, Sigmoid

**Decoder** (for $\boldsymbol{\sigma}_t^+$ or $\sigma_t^+$): 1 fully connected + (elu + 1):

- Fully Connected 1: 10, ReLU

### D.3 KITTI

**Observation and Data Set:** Images from the KITTI (Geiger et al., 2012) dataset for visual odometry. Sequences 00, 01, 02, 08, 09 used for training. Sequences 03, 04, 05, 06, 07, 10 for testing.
**Encooder**: FlowNet2 + 4 Convolutional + 2 Branches of dense Layers. See (Zhao et al., 2018)
**Transition Model:** 100 dimensional latent observations, 200 dimensional latent state, bandwidth: 5, number of basis: 20

- $\alpha(\mathbf{z}_t)$: No hidden layers - softmax output

**Decoder** (for $\mathbf{s}_t^+$): 2 fully connected + linear output:

- Fully Connected 1: 128, ReLU
- Fully Connected 1: 128, ReLU

**Decoder** (for $\boldsymbol{\sigma}_t^+$ or $\sigma_t^+$): 2 fully connected + (elu + 1):

- Fully Connected 1: 128, ReLU
- Fully Connected 1: 128, ReLU

### E OBSERVATION NOISE GENERATION PROCESS

Let $\mathcal{U}(x, y)$ denote the uniform distribution from $x$ to $y$. To generate the noise for the pendulum task for each sequence a sequence of factors $f_t$ of same length was generated. To correlate the factors they were sampled as $f_0 \sim \mathcal{U}(0, 1)$ and $f_{t+1} = \min(\max(0, f_t + r_t), 1)$ with $r_t \sim \mathcal{U}(-0.2, 0.2)$. Afterwards, for each sequence two thresholds $t_1 \sim \mathcal{U}(0.0, 0.25)$ and $t_2 \sim \mathcal{U}(0.75, 1)$ were sampled.

All $f_t < t_1$ were set to 0, all $f_t > t_2$ to 1 and the rest was linearly mapped to the interval $[0, 1]$. Finally, for each image $\mathbf{i}_t$ an image consisting of pure uniformly distributed noise $\mathbf{i}_t^{noise}$ was sampled and the observation computed as $\mathbf{o}_t = f_t \cdot \mathbf{i}_t + (1 - f_t) \cdot \mathbf{i}_t^{noise}$.

