# OpenReview forum: "Recurrent Kalman Networks: Factorized Inference in High-Dimensional Deep Feature Spaces"
_ICLR.cc/2019/Conference_

### Official Review · AnonReviewer1 · 2018-11-03
**Nice work but misses some significant work from the literature**

**Rating:** 6
**Confidence:** 4

**Review:**

This paper proposes, Recurrent Kalman Network, a modified Kaman filter in which the latent dynamics is projected into a higher dimensional space; efficient inference in this high-dimensional latent space is possible due to the space being locally linear. The state representation, transition, and observation models are learned jointly by backpropagation.
The paper is well written and the model is clearly explained; I also like the simplicity of the idea that uses the same machinery of Kalman filter.  However, I believe the authors can improve the presentation of the model and empirical evaluation.

In terms of model presentation, the authors can compare the model with a large set of deep recurrent models that have recently been proposed for modeling time series with nonlinear latent dynamics (e.g. Variational Sequential Monte Carlo, Structured inference networks for nonlinear state space models, Black box variational inference for state space models, Composing graphical models with neural networks for structured representations and fast inference, etc.). For instance, a table of some of these models with their pros and cons can be helpful for guiding the reader.

In terms of model evaluation, the paper needs a better evaluation section specifically on the generative models (see examples above) that are much  more suitable for modeling uncertainty compared to LSTM/GRU. More specifically, another approach for alleviating the limitations of Kalman filter would be to use non-linear transitions based on some non-linear functions approximation. This approach has been proposed in deep Kaman filter (Krishnan 2015) and it would be interesting to see how well your model performs compared to that for modeling uncertainty and computing predictive log-likelihood.

In conclusion, I think the paper presents a nice idea but it requires more work in order to pass the ICLR acceptance threshold.

----------------------------------------------------------
The authors have addressed my comments and as a result I changed my rating to 6.

---

> ### Author Response · Authors · 2018-11-22
> **Paper revised based on reviewers comments**
>
> Thank you for your time and valuable feedback that helped us to improve the paper considerably. We implemented the following improvements to the paper based on the comments of the reviewers:
>
> 1.) We added a qualitative comparison to recent probabilistic generative model approaches that also use KF related techniques, see Table 1. This analysis shows that our approach is more general while allowing for more simple learning methods than other probabilistic SOTA methods.
>
> 2.) We explicitly stated the update equations used by the RKN (see section 2.4) and clarified Figure 1 to better visualize the general structure.
>
> 3.) We explicitly state the likelihood objective in section 2.5
>
> 4.) We added a quantitative comparison to Embed to Control (Watter et. al 2015), Structured Inference Networks (Krishnan et al. 2017) and the Kalman Variational Autoencoder (Fraccaro et al. 2017) on the image imputation tasks for the pendulum and the quadlink. See Table 3 and section 3.3. Those experiments show that our approach outperforms the other models despite using a much easier optimization scheme and less information (smoothing vs. filtering).
>
> 5.) Reedited abstract, introduction and related work to clarify the contribution of the paper in relation to recent SOTA.
>
> For more specific answers to the comments, please see below:
>
> “In terms of model presentation… “ - We reworked the related work section and added a table comparing our approach to a variety of recent works on a qualitative level. This comparison shows while most generative modelling approaches have been used to predict future or missing images, they can not be used directly for state estimation. Our model is more general and can be used straightforwardly for image prediction as well as for state estimation. Moreover, our model does not require the use of approximate inference methods such as variational inference but can be learned directly in an end to end manner. We believe that this is one of the main reasons why the RKN outperforms all other generative models in the imputation experiment that is now added in Section 3.
>
> “In terms of model evaluation” -  We added a quantitative comparison in Section 3 where we compare our approach to different generative models from the literature. The results show that while the RKN is using a much simpler model and/or learning method than related approaches, we significantly outperform them on the image imputation task in terms of the log-likelihood loss function. Note that we had to switch from a Gaussian to a Bernoulli distribution for creating images to make this comparison easier

---

### Official Review · AnonReviewer2 · 2018-11-03
**Interesting idea, but insufficient comparison to existing work**

**Rating:** 6
**Confidence:** 3

**Review:**

PAPER SUMMARY
-------------
This paper proposes a method for inferring the latent state and making predictions based on a sequence of observations. The idea is to map the observation to a latent space where the relation to the latent state is linear, and the dynamics of the latent state are locally linear. Therefore, in this latent space a Kalman filter can be applied to infer the current state and predict the next state, including uncertainty estimates. Finally, the predicted latent state is mapped to a prediction for the observation or some other variable of interest.

The experiments show that the proposed approach slightly outperform LSTM and a GRU based approaches.


POSITIVE ASPECTS
----------------
- The idea of applying a Kalman filter in a latent space is interesting.
- The experimental results show that the proposed approach outperforms LSTM and a GRU based approaches.
- The paper is well written.

NEGATIVE ASPECTS
----------------
- The observation noise sigma^obs is a function of the observation itself. This seems strange, since typically the observation does not contain itself the information about how much it has been corrupted by noise. This choice should be discussed in more detail, especially what kind of assumptions this implies about the underlying process.
- I believe that a more detailed comparison to existing approaches finding a latent space from a sequence of observations would be necessary, both on a technical as well as on an experimental level. For instance, a technical comparison to the approach from Watter et al. 2015 would be appropriate, since it is similar in the sense that the latent space is optimized to have locally linear dynamics.
Furthermore, an experimental comparison to Watter et al. 2015 and [1] would be relevant.



[1] Wahlström et al. 2015 - From Pixels to Torques - Policy Learning with Deep Dynamical Models

---

> ### Author Response · Authors · 2018-11-22
> **Paper revised based on reviewers comments**
>
> Thank you for your time and valuable feedback that helped us to improve the paper considerably. We implemented the following improvements to the paper based on the comments of the reviewers:
>
> 1.) We added a qualitative comparison to recent probabilistic generative model approaches that also use KF related techniques, see Table 1. This analysis shows that our approach is more general while allowing for more simple learning methods than other probabilistic SOTA methods.
>
> 2.) We explicitly stated the update equations used by the RKN (see section 2.4) and clarified Figure 1 to better visualize the general structure.
>
> 3.) We explicitly state the likelihood objective in section 2.5
>
> 4.) We added a quantitative comparison to Embed to Control (Watter et. al 2015), Structured Inference Networks (Krishnan et al. 2017) and the Kalman Variational Autoencoder (Fraccaro et al. 2017) on the image imputation tasks for the pendulum and the quadlink. See Table 3 and section 3.3. Those experiments show that our approach outperforms the other models despite using a much easier optimization scheme and less information (smoothing vs. filtering).
>
> 5.) Reedited abstract, introduction and related work to clarify the contribution of the paper in relation to recent SOTA.
>
> For more specific answers to the comments, please see below:
>
> “The observation noise sigma^obs is a function of the observation itself. This seems strange” - For high dimensional observations such as images, the amount of noise can often be inferred from the observation itself. For example, if certain relevant aspects of the scenario are occluded in an image this is useful information which can be extracted from the image.
> Hence, making the variance depend on the images is quite common. It is done in the  approaches using variational autoencoder we compared to (Watter et al. 2015, Karl et al. 2016, Fraccaro et al. 2017) and was also a crucial aspect of the BackpropKF (Haarnoja et al. 2016)
>
> In fact, making sigma^obs dependent on the observation is crucial for our approach since it allows the encoder to express the uncertainty of its current estimate and allows the model to ignore the current observation if it is not useful. This property is needed in all our experiments.
>
> “I believe that a more detailed...” - We reworked the related work section, added a table comparing our approach to a variety of recent work including probabilistic generative models on a qualitative level. The main result of this qualitative comparison is that, while our model is more general than most of the current SOTA approaches as it can be used straightforwardly for state estimation as well as image prediction, it also offers the simplest training method (end-to-end training by backpropagation) without the need of approximate inference methods such as variational inference, that is required by most probabilistic generative models.
> We also added a quantitative comparison in Section 3 on the image imputation task that allows a direct comparison to the probabilistic generative models. While our approach offers a much simpler and more direct learning approach, it outperforms more complex models considerably on this task.  Note that we had to switch from a Gaussian to a Bernoulli distribution for generating images to make this comparison easier.

---

### Official Review · AnonReviewer3 · 2018-11-05
**Interesting model needs more context**

**Rating:** 6
**Confidence:** 4

**Review:**

This paper presents a particular architecture for a probabilistic recurrent neural network that is based on ideas from Kalman filtering. Whereas Kalman filters are used to infer the state of a known generative model (a linear-Gaussian dynamical system), here, the authors jointly learn a recursive filter without explicitly formulating a generative model of the data.

The paper deals with an important problem and the approach has many appealing characteristics: it learns a state representation and its associated transition dynamics, it learns nonlinear filter that can be used online and it learns encoders/decoders from high-dimensional observations to the state.

The article does not provide any probability density (even though learning happens by maximizing a likelihood) and there are no connections to probabilistic generative models. In my opinion this is a pity since this would shed more light into the characteristics of the proposed approach.

I believe that the model could be presented more clearly. For example, the Preliminaries section uses formulas before defining them. Also, explicitly writing the high-level chain of computations from o_t and z_{t-1}^+ to o_t^+ and s_t^+ would be extremely useful. Even more than Fig. 1, in my opinion.

All in all, I have found this an interesting architecture for a RNN but would have appreciated more insight into its relationships with the large body of generative probabilistic state-space models and the methods to perform inference on them.

---

> ### Author Response · Authors · 2018-11-22
> **Paper revised based on reviewers comments**
>
> Thank you for your time and valuable feedback that helped us to improve the paper considerably. We implemented the following improvements to the paper based on the comments of the reviewers:
>
> 1.) We added a qualitative comparison to recent probabilistic generative model approaches that also use KF related techniques, see Table 1. This analysis shows that our approach is more general while allowing for more simple learning methods than other probabilistic SOTA methods.
>
> 2.) We explicitly stated the update equations used by the RKN (see section 2.4) and clarified Figure 1 to better visualize the general structure.
>
> 3.) We explicitly state the likelihood objective in section 2.5
>
> 4.) We added a quantitative comparison to Embed to Control (Watter et. al 2015), Structured Inference Networks (Krishnan et al. 2017) and the Kalman Variational Autoencoder (Fraccaro et al. 2017) on the image imputation tasks for the pendulum and the quadlink. See Table 3 and section 3.3. Those experiments show that our approach outperforms the other models despite using a much easier optimization scheme and less information (smoothing vs. filtering).
>
> 5.) Reedited abstract, introduction and related work to clarify the contribution of the paper in relation to recent SOTA.
>
> For more specific answers to the comments, please see below:
>
> “The article does not provide any probability density ..” - We stated the likelihood objective more explicitly in section 2.5.
>
>  “...and there are no connections to probabilistic generative models.” - We reworked the related work section, added a table comparing our approach to a variety of recent work including probabilistic generative models on a qualitative level. The main result of this qualitative comparison is that, while our model is more general than most of the current SOTA approaches as it can be used straightforwardly for state estimation as well as image prediction, it also offers the simplest training method (end-to-end training by backpropagation) without the need of approximate inference methods such as variational inference, that is required by most probabilistic generative models.
> We also added a quantitative comparison in Section 3 on the image imputation task that allows a direct comparison to the probabilistic generative models. While our approach offers a much simpler and more direct learning approach, it outperforms more complex models considerably on this task.  Note that we had to switch from a Gaussian to a Bernoulli distribution for generating images to make this comparison easier.
>
> “the Preliminaries section uses formulas before defining them” - We fixed that.
>
> “Also, explicitly writing...”  - We reworked Figure 1 and moved some elaboration on the equations from the appendix to the main part (specifically, section 2.4). We hope that clarifies our presentation.

---

### Meta-Review · Area_Chair1 · 2018-12-14
**Borderline - but missing clarity**

**Confidence:** 4
**Recommendation:** Reject

**Metareview:**

A lot of work has appeared recently on recurrent state space models. So although this paper is in general considered favorable by the reviewers it is unclear exactly how the paper places itself in that (crowded) space. So rejection with a strong encouragement to update and resubmission is encouraged.